# Functional evolution of a morphogenetic gradient

Chun Wai Kwan[1], Jackie Gavin-Smyth[2], Edwin L Ferguson[1,3], Urs Schmidt-Ott[1]*

[1]Department of Organismal Biology and Anatomy, University of Chicago, Chicago, United States; [2]Department of Ecology and Evolution, University of Chicago, Chicago, United States; [3]Department of Molecular Genetics and Cell Biology, University of Chicago, Chicago, United States

**Abstract** Bone Morphogenetic Proteins (BMPs) pattern the dorsal-ventral axis of bilaterian embryos; however, their roles in the evolution of body plan are largely unknown. We examined their functional evolution in fly embryos. BMP signaling specifies two extraembryonic tissues, the serosa and amnion, in basal-branching flies such as *Megaselia abdita*, but only one, the amnioserosa, in *Drosophila melanogaster*. The BMP signaling dynamics are similar in both species until the beginning of gastrulation, when BMP signaling broadens and intensifies at the edge of the germ rudiment in *Megaselia*, while remaining static in *Drosophila*. Here we show that the differences in gradient dynamics and tissue specification result from evolutionary changes in the gene regulatory network that controls the activity of a positive feedback circuit on BMP signaling, involving the *tumor necrosis factor alpha* homolog *eiger*. These data illustrate an evolutionary mechanism by which spatiotemporal changes in morphogen gradients can guide tissue complexity.

## Introduction

The specification of different cell fates by morphogen gradients has been a longstanding focus within developmental biology. While it is well established that gradients of diffusible morphogens produce complex pattern during development, their role as drivers of morphological evolution has mostly been inferred from theoretical studies, due to the challenge of quantifying and functionally assessing their activities in species outside of select genetic model organisms (*Turing, 1952*; *Kondo and Miura, 2010*; *Green and Sharpe, 2015*; *Marcon et al., 2016*). Bone Morphogenetic Proteins (BMPs) pattern the embryonic dorsal-ventral axis of bilaterian embryos, raising the question of the role of the BMP gradient in the evolution of body plans (*Bier and De Robertis, 2015*). To address this question, we compared the functions of embryonic BMP gradients in two fly species that differ in tissue complexity downstream of BMP signaling.

In *Drosophila melanogaster* embryos, the BMP gradient forms through directed extracellular BMP movement and initiates a positive feedback circuit leading to a bistable pattern of BMP signaling by the end of the blastoderm stage (reviewed in *O'Connor et al., 2006*; *Shilo et al., 2013*; *Wharton and Serpe, 2013*). High levels of BMP signaling, centered on the dorsal midline, specify a single extraembryonic tissue, the amnioserosa. However, in basal-branching flies, including *Megaselia abdita* (Phoridae), BMP signaling specifies two extraembryonic tissues, the serosa and the amnion (*Schmidt-Ott and Kwan, 2016*). Previously, we showed that the dynamics of BMP signaling in the blastoderm are similar between *Megaselia* and *Drosophila*, but differ in the early gastrula when the *Megaselia* gradient broadens while the *Drosophila* gradient remains static (*Rafiqi et al., 2012*). Here, we show that differences in the control of a positive feedback circuit involving *eiger* (*egr*) (*Gavin-Smyth et al., 2013*) are responsible for the altered dynamics of BMP signaling and amnion

*For correspondence: uschmid@
uchicago.edu

**Competing interests:** The authors declare that no competing interests exist.

specification in *Megaselia*. We hereby reveal an evolutionary mechanism by which morphogen gradients can alter the complexity of tissue types between species.

## Results and discussion

### BMP signaling during gastrulation is necessary and sufficient for amnion specification

In *Megaselia* and *Drosophila*, BMP signaling specifies extraembryonic membranes (*Figure 1A*) and can be quantified by staining with an antibody specific to the activated phosphorylated form of Mad (pMad), an essential transcriptional effector of the BMP pathway (*Dorfman and Shilo, 2001*). During early blastoderm stages in both species BMP signaling is initially low and broadly distributed over the dorsal regions of the embryo but refines into a narrow dorsal stripe of high activity by the onset of gastrulation. However, during early gastrulation in *Megaselia*, the BMP signaling domain broadens to encompass the edge of the germ rudiment comprising the presumptive amnion, while the domain in *Drosophila* remains static (*Figure 1B*).

In *Megaselia*, serosa and amnion specification can be visualized with a combination of genetic markers. A homolog of *zerknüllt* (*Mab-zen*), which encodes a homeodomain protein, marks and specifies serosa cells in blastoderm (stage 5) and gastrula embryos (stage 6) (*Rafiqi et al., 2008*). Homologs of *hindsight* (*Mab-hnt*) and *dorsocross* (*Mab-doc, Mab-doc2*) (*Figure 2A* and *Figure 2—figure supplement 1*), which encode zinc-finger and T-box proteins respectively, are also expressed in stage 5 and 6 embryos in a slightly wider domain than *Mab-zen* (*Figure 2—figure supplement 2*) (*Rafiqi et al., 2012*), encompassing both the prospective serosa and amnion. Lastly, a homolog of the TNF alpha gene *eiger* (*Mab-egr*) is expressed in the serosa and amnion of gastrulating *Megaselia* embryos (*Figure 2B–C*). *Mab-egr* expression continues until dorsal closure, but from germ band extension (stage 11) onwards it is expressed only in the amnion cells, which at this stage are polyploid and much larger than the adjacent embryonic cells (*Figure 2D–E* and *Figure 2—figure*

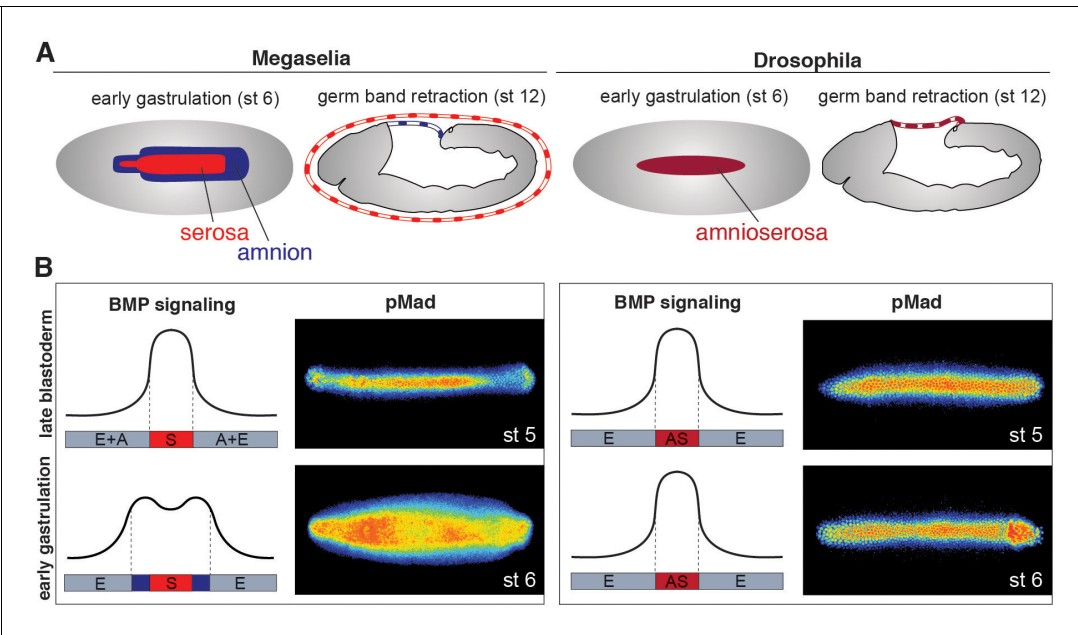

**Figure 1.** Extraembryonic tissue and BMP signaling differ between *Megaselia* and *Drosophila*. (**A**) Schematics of *Megaselia* embryos with serosa and amnion and of *Drosophila* embryos with amnioserosa at the beginning of gastrulation (stage 6, left) and during early germ band retraction (stage 12, right), modified from *Rafiqi et al. (2012)*. Here, as in all subsequent figures, blastoderm and gastrula stages are shown in dorsal view while later stages are shown in lateral view with the dorsal side up unless specified otherwise. Anterior is always left. (**B**) Schematic pMad intensity profiles at the dorsal midline relative to prospective serosa (S, red), amnion (A, blue), amnioserosa (AS, maroon), and embryonic tissues (E, grey) in *Megaselia* and *Drosophila*. Representative embryos stained for pMad on right.

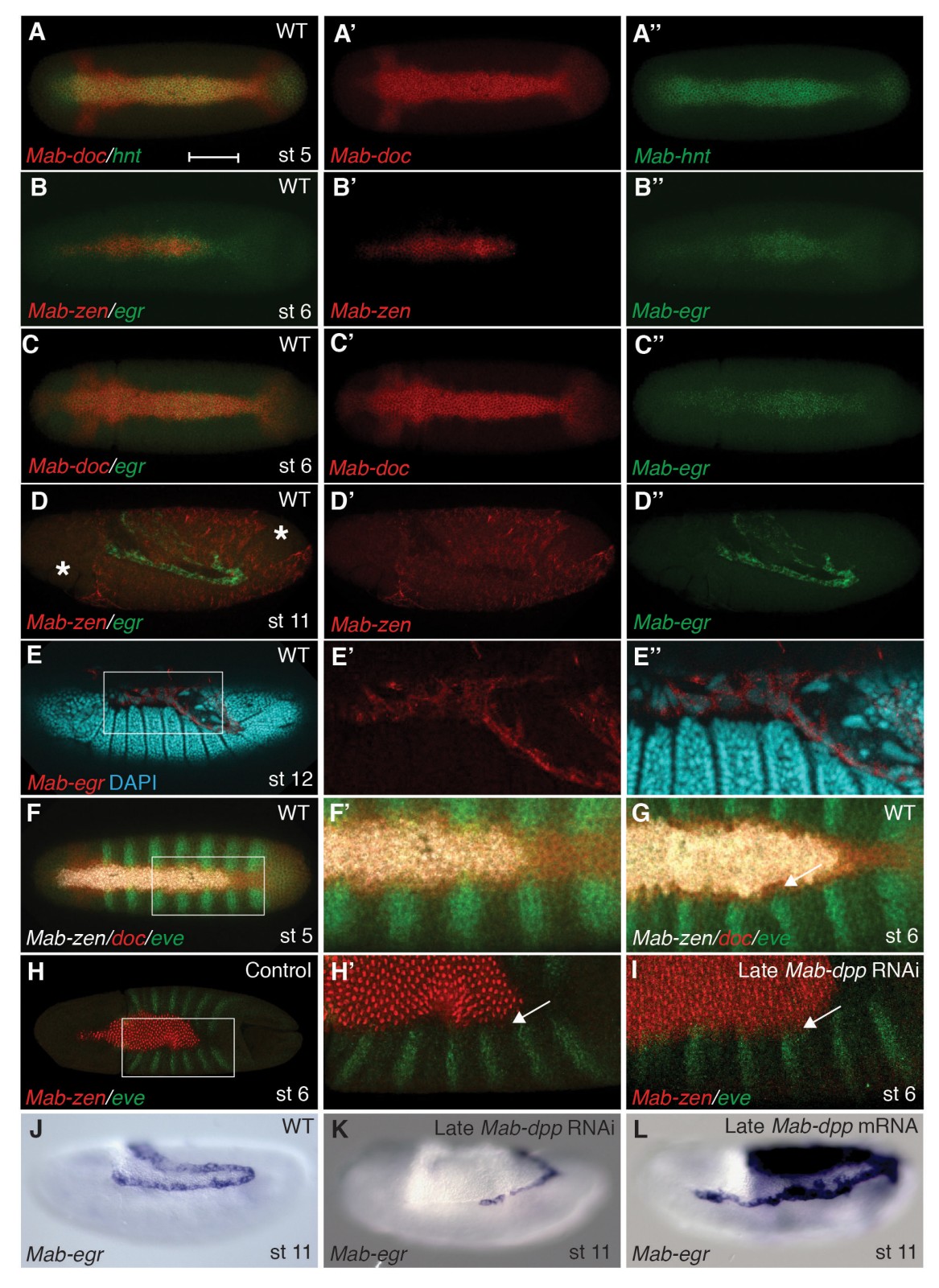

**Figure 2.** Specification of amnion by BMP signaling in *Megaselia*. (A) *Mab-hnt* and *Mab-doc* expression at the late blastoderm stage. Scale bar = 100 μm. (B, C) *Mab-egr* and *Mab-zen* (B) and *Mab-egr* and *Mab-doc* (C) expression at early gastrulation. (D) *Mab-egr* and *Mab-zen* expression after germ band extension. Asterisks denote tears in the serosa during sample preparation. (E) *Mab-egr* expression during germ band retraction. The serosa has been removed and nuclei have been labeled with DAPI. Boxed region enlarged (E'–E''). (F, G) *Mab-doc, Mab-zen* and *Mab-eve* expression at late

*Figure 2 continued on next page*

Figure 2 continued

blastoderm stage (stage 5) (F, enlargement F') and early gastrulation (stage 6) (G) with arrow pointing to abutting *Mab-eve* and *Mab-doc* expression domains. (H, I) *Mab-zen* and *Mab-eve* expression in early gastrula control embryo (H, enlargement H') and following *Mab-dpp* knockdown after 50% blastoderm cellularization (I). Arrows, gap between the *Mab-eve* and *Mab-zen* domains (H') that is suppressed in the knockdown embryo (I). (J–L) *Mab-egr* expression at germ band extension in wild-type embryo (J), and after *Mab-dpp* knockdown (K) or *Mab-dpp* overexpression (L) after 50% blastoderm cellularization.

The following figure supplements are available for figure 2:

**Figure supplement 1.** Expression of *Mab-doc2.*

**Figure supplement 2.** Time course of *Mab-zen*, *Mab-doc* and *Mab-eve* expression.

**Figure supplement 3.** .Expression of *Mab-egr.*

**Figure supplement 4.** Overexpression of *Mab-eve* represses *Mab-zen* expression.

supplement 3). The time course of serosa and amnion specification is suggested by the dorsal repression of an embryonic marker, *Megaselia even-skipped* (*Mab-eve*) (*Rafiqi et al., 2012*). In wild-type blastoderm embryos, the repression of *Mab-eve* extends laterally from the dorsal midline to the boundary of the *Mab-zen* domain, but after gastrulation begins *Mab-eve* expression withdraws further to abut the *Mab-doc/hnt* domain (*Figure 2F–G* and *Figure 2—figure supplement 2*), likely as a result of repression by BMP signaling. Conversely, overexpression of *Mab-eve* suppresses *Mab-zen* expression (*Figure 2—figure supplement 4*). Thus, amnion specification might not be completed before the onset of gastrulation. At stage 6, the amnion anlage of *Megaselia* is defined as the thin band of cells expressing *Mab-doc/doc2*, *Mab-hnt*, and *Mab-egr* but not *Mab-zen* while, after germ band extension, mature amnion cells are defined by their large size and *Mab-egr* expression.

To test the hypothesis that temporally distinct BMP signaling sequentially specifies the two extra-embryonic membranes, we decreased BMP signaling during gastrulation by *Mab-dpp* knockdown after 50% blastoderm cellularization and monitored the expression of *Mab-eve*. Knockdown of *Mab-dpp* by injection of double stranded RNA (dsRNA) at the early blastoderm stage completely suppresses both serosa and amnion specification and results in the circumferential expression of *Mab-eve* (*Rafiqi et al., 2012*). In contrast, in all 13 late *Mab-dpp* knockdown embryos fixed during early gastrulation, the expression of *Mab-zen* was not affected; however, in five embryos the repression of *Mab-eve* in the amnion anlage was incomplete (*Figure 2H–I*). Late *Mab-dpp* knockdown reduced *Mab-egr* expression in a majority of stage 11/12 embryos (55%, n = 40) whereas late ectopic *Mab-dpp* expression caused an expansion of the *Mab-egr* domain in at least 35% of the embryos (n = 57) (*Figure 2J–L*). Taken together, these data provide evidence that BMP signaling during gastrulation is necessary and sufficient for amnion specification.

### *Mab-doc* promotes amnion development by elevating BMP signaling in the amnion anlage at the onset of gastrulation

In *Drosophila*, both BMP signaling and *zen* are necessary at the blastoderm stage for the expression of the three *doc* paralogs and *hnt* in the amnioserosa anlage, although the essential function of these genes in amnioserosa maintenance becomes apparent only after gastrulation (*Yip et al., 1997*; *Reim et al., 2003*). However, in *Megaselia*, BMP signaling activates *Mab-doc* and *Mab-hnt* independently from *Mab-zen* (*Figure 3—figure supplement 1A–I*), suggesting these genes could play a role in amnion specification. Following knockdown of *Mab-doc/doc2* or *Mab-hnt* activity by RNAi, we observed confluent expression domains of *Mab-zen* and *Mab-eve* during early gastrulation (4/9 and 5/11 embryos, respectively; *Figure 3A–C*), and a decrease in *Mab-egr* expression at stage 11/12 (*Figure 3D–F*). Knockdown of the activities of all three genes eliminated *Mab-egr* expression in germ-band extended embryos (*Figure 3D*), indicating that *Mab-doc/doc2* and *Mab-hnt* together are essential for amnion specification in *Megaselia*. Conversely, overexpression of *Mab-doc* or *Mab-hnt* induced ectopic amnion, as evidenced by an enlargement of the *Mab-egr* domain at stage 11/12

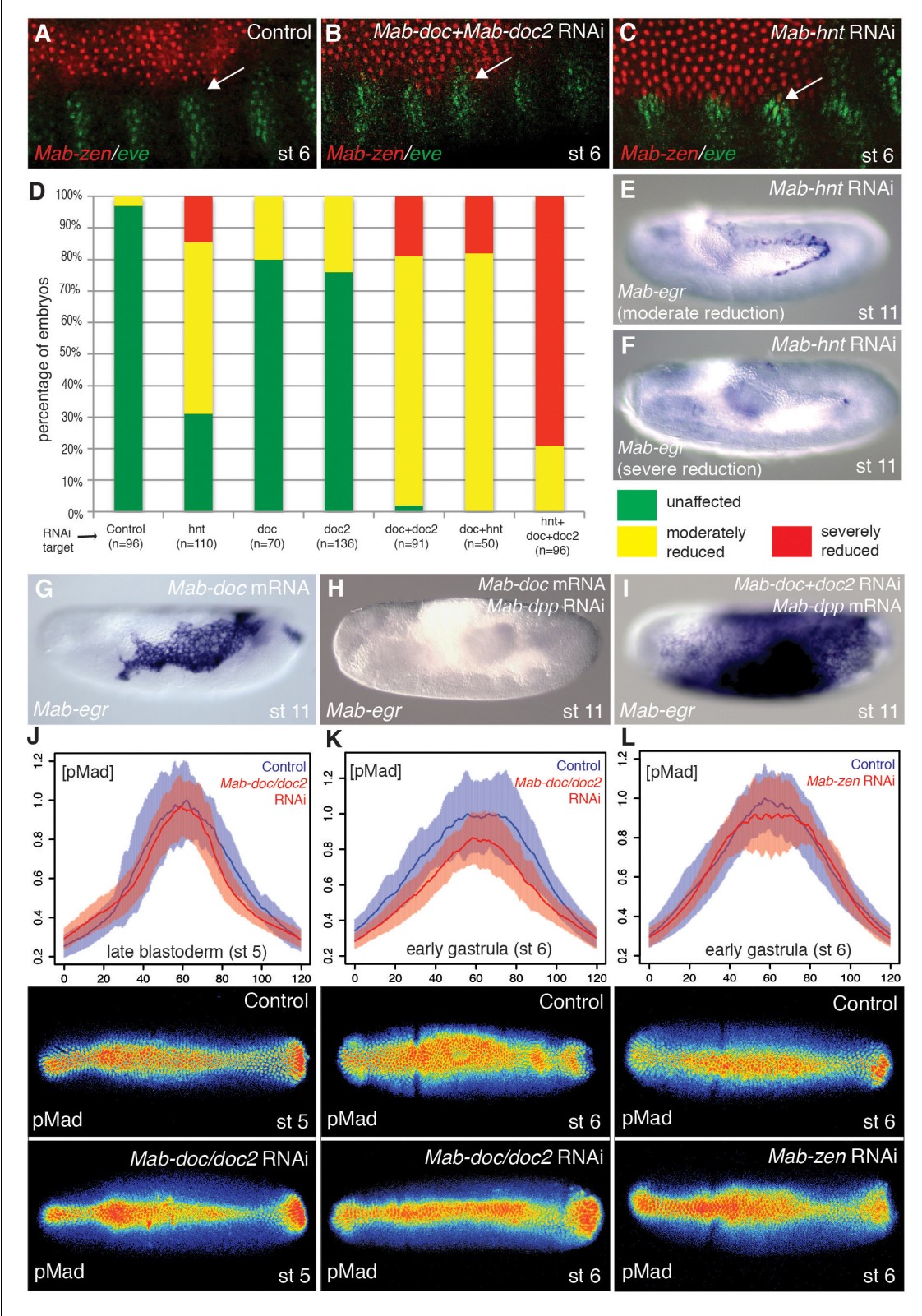

**Figure 3.** *Mab-doc* and *Mab-hnt* elevate BMP signaling to specify amnion in *Megaselia*. (A–C) *Mab-zen* and *Mab-eve* expression in early gastrula control embryo (A) and after *Mab-doc/doc2* knockdown (B) or *Mab-hnt* knockdown (C). Arrows, gap between the *Mab-eve* and *Mab-zen* domains (A) that is suppressed in the knockdown embryos (B, C). (D–F) Bar chart (D) quantifying the reduction of *Mab-egr* expression at stage 11/12 after *Mab-hnt* and/or *Mab-doc/doc2* knockdown, and representative embryos of moderately reduced (yellow, E), or severely reduced (red, F) phenotypes. (G–I) *Mab-*

*Figure 3 continued on next page*

*Figure 3 continued*

*egr* expression at germ band extension following *Mab-doc* overexpression (**G**), *Mab-doc* overexpression and *Mab-dpp* knockdown (**H**), and *Mab-dpp* overexpression and *Mab-doc/doc2* knockdown (**I**). (**J–L**) Mean and shaded standard deviation of pMad intensities in control injected embryos (blue) and in *Mab-doc/doc2* knockdown embryos (red) at the cellular blastoderm stage (n = 10, control n = 10) (**J**), at early gastrulation (n = 11, control n = 11) (**K**), and in *Mab-zen* knockdown embryos (red) at early gastrulation (n = 10, control n = 17) (**L**) with representative embryos stained for pMad underneath each plot.

The following figure supplement is available for figure 3:

**Figure supplement 1.** Function and regulation of *Mab-doc* and *Mab-hnt.*

---

(*Figure 3G* and *Figure 3—figure supplement 1J–L*). Overexpression of *Mab-doc* could bypass the requirement for *Mab-hnt*, while overexpression of *Mab-hnt* could not bypass the requirement for *Mab-doc* (*Figure 3—figure supplement 1M–N*), consistent with the hypothesis that *Mab-doc* and *Mab-hnt* share a common target necessary for amnion formation that is primarily dependent upon *doc* activity.

The activities of *Mab-doc* and *Mab-hnt* could promote the amnion formation in an instructive manner, by activating the amnion gene network of *Megaselia*, or they might promote the amnion formation in a permissive manner, e.g., by elevating BMP signaling. Overexpression of *Mab-doc* in *Mab-dpp* knockdown embryos did not induce any expression of *Mab-egr* in stage 11/12 embryos (n = 44) (*Figure 3H*). Conversely, overexpression of *Mab-dpp* in *Mab-doc/doc2* knockdown embryos resulted in ectopic expression of *Mab-egr* at stage 11/12 (36%, n = 47) (*Figure 3I*). Thus, BMP signaling is sufficient to direct the expression of amnion specific genes in the absence of *Mab-doc/doc2* activity. To confirm that this result was not due to an excessive non-physiological level of Mab-Dpp produced by the injected mRNA, we asked whether the endogenous level of BMP signaling at the dorsal midline in the blastoderm embryo would be sufficient to specify amnion in the absence of both *Mab-doc/doc2* and the serosal determinant *Mab-zen*. Knockdown of *Mab-zen* partially restored amnion in *Mab-doc/doc2* knockdown embryos (*Figure 3—figure supplement 1O*). Taken together, these data suggest *Mab-doc* promotes amnion formation permissively.

To directly test whether *Mab-doc* activity elevates BMP signaling, we quantified the intensity of pMad staining in embryos after *Mab-doc/doc2* knockdown. While *Mab-doc/doc2* knockdown had little effect on pMad levels during the late blastoderm stage compared to control embryos (Wilcoxon rank sum test, p=0.3697; *Figure 3J*), in early gastrula stage embryos, knockdown of *Mab-doc/doc2* resulted in significantly reduced pMad levels compared to control embryos (Wilcoxon rank sum test, p=0.01165; *Figure 3K*). In contrast, knockdown of *Mab-zen* did not alter the average level of pMad at the beginning of gastrulation (Wilcoxon rank sum test, p=0.2367; *Figure 3L*). The observation that *Mab-doc/doc2* is dispensable for amnion cell fate specification but necessary for wild-type levels of BMP signaling at the early gastrula stage strongly support the model that amnion formation is driven by a *Mab-doc*-dependent elevation of BMP signaling in the amnion anlage at the onset of gastrulation.

### *Mab-doc*-dependent control of *Mab-egr* expression contributes to a positive feedback circuit that promotes BMP signaling during gastrulation

We then explored the mechanism by which *Mab-doc* promotes BMP signaling at the gastrula stage. Embryos injected centrally with *Mab-doc* mRNA displayed a local expansion of the pMad domain during gastrulation (15/15) coupled with the frequent depletion of endogenous pMad in adjacent regions (12/15) (*Figure 4A*). This result parallels a phenotype observed in *Drosophila* where injection of mRNA encoding activated BMP receptors into the blastoderm embryo causes an increase in BMP ligand-receptor interactions, coupled with a decrease in BMP ligand-receptor binding in nearby regions (*Wang and Ferguson, 2005*). These data indicate that a positive feedback circuit downstream of BMP signaling increases local receptor-ligand interactions, and that, due to a limiting amount of BMP ligand, ligand-receptor interactions decrease in nearby regions. Conversely, *Megaselia* embryos injected with *Mab-zen* mRNA (n = 11) had a similar pMad domain to injected control embryos (n = 12) and developed a reduced or abnormal amnion (44/51) (*Figure 4—figure*

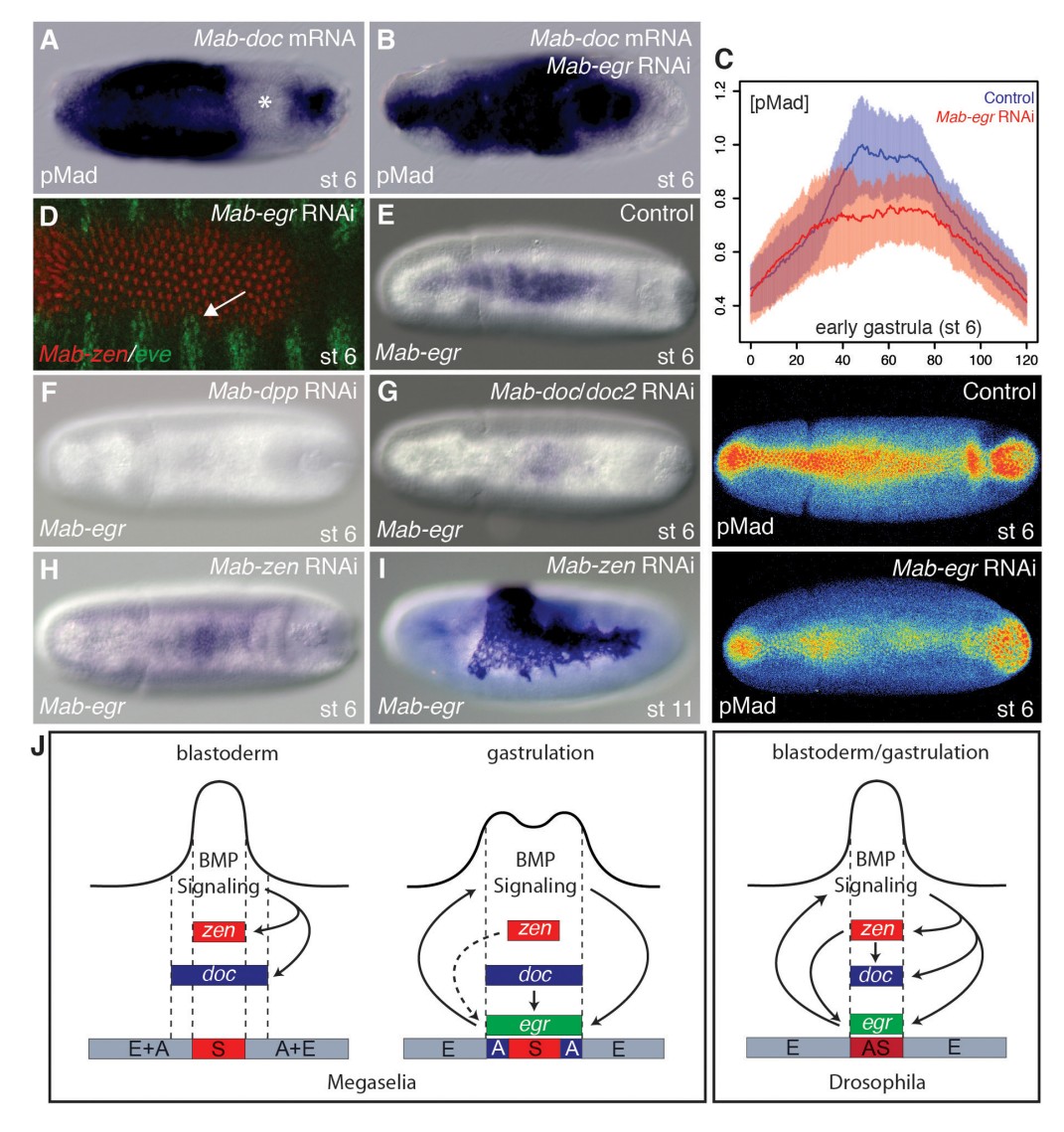

**Figure 4.** *Mab-egr* is downstream of *Mab-doc* and promotes BMP signaling. (A, B) pMad staining following *Mab-doc* overexpression (A) or *Mab-doc* overexpression and *Mab-egr* knockdown (B). The asterisk marks site of endogenous pMad depletion. (C) Mean intensity and standard deviation of pMad staining in control injected embryos (blue, n = 10) and *Mab-egr* knockdown embryos (red, n = 9) at early gastrulation with representative embryos stained for pMad underneath the plot. (D) *Mab-zen* and *Mab-eve* expression at early gastrulation after *Mab-egr* knockdown with arrow indicating suppressed gap between the *Mab-eve* and *Mab-zen* domains. (E–H) *Mab-egr* expression in wild type (E), *Mab-dpp* knockdown (F), *Mab-doc/doc2* knockdown (G), and *Mab-zen* knockdown (H) embryos at early gastrulation. (I) *Mab-egr* expression in a *Mab-zen* knockdown embryo after germ band extension. (J) BMP gene regulatory networks in *Megaselia abdita* and *Drosophila melanogaster*.

The following figure supplements are available for figure 4:

**Figure supplement 1.** Effect of *Mab-zen* overexpression on BMP signaling.
**Figure supplement 2.** Efficiency of *Mab-egr* RNAi.
**Figure supplement 3.** Expression profile of *Mab-cv-2*.
**Figure supplement 4.** *Mab-cv-2* promotes amnion specification.
**Figure supplement 5.** Regulation of *Mab-cv-2* is largely independent of *Mab-doc/doc2*.

supplement 1). These results suggest *Mab-doc*, but not *Mab-zen*, locally activates a positive feed-back circuit in the *Megaselia* embryo, where BMP ligands are limiting.

Recent experiments in *Drosophila* identified *egr* activity as a component of the positive feedback circuit (*Gavin-Smyth et al., 2013*). To determine whether *Mab-egr* could be a component of a positive feedback circuit in *Megaselia*, we asked whether knockdown of *Mab-egr* could modify the phenotype caused by injection of *Mab-doc* mRNA. While the pMad domain in all *Mab-doc* injected embryos was locally expanded (14/14) (*Figure 4B*), only a few embryos (2/14) displayed a depletion of endogenous pMad in adjacent regions. These data indicate that *Mab-egr* increases the ability of cells overexpressing *Mab-doc* to compete for BMP ligands during early gastrulation.

In *Drosophila*, loss of *egr* reduces intensity of pMad staining by 50% (*Gavin-Smyth et al., 2013*). Similarly, we found that, at the onset of gastrulation, pMad levels in *Mab-egr* knockdown embryos were reduced by about 50% on average (Wilcoxon rank sum test, p=0.00381) (*Figure 4C* and *Figure 4—figure supplement 2*). Moreover, confluent expression domains of *Mab-eve* and *Mab-zen* could be observed in such embryos (3/10) (*Figure 4D*). As *Mab-egr* expression extends to the edge of the gastrulating germ rudiment, these observations suggest that *Mab-egr* promotes amnion specification downstream of *Mab-doc/doc2* by elevating BMP signaling during gastrulation in prospective amnion cells.

The pMad gradients of *Mab-egr* RNAi embryos were on average slightly broader than in *Mab-doc/doc2* RNAi embryos (*Figures 3K* and *4C*), suggesting that *Mab-doc/doc2* might control more than one gene with a role in shaping the BMP gradient, similar to *Drosophila* where the BMP-dependent feedback circuit appears to involve more genes than just *egr* (*Gavin-Smyth et al., 2013*). As a potential second *Mab-doc/doc2* target we examined the *Megaselia* ortholog of *cv-2* (*Figure 4—figure supplement 3*), which encodes an extracellular, context- and concentration-dependent modulator of BMP signaling (*Conley et al., 2000*; *Serpe et al., 2008*). However, although *Mab-cv-2* likely also promotes BMP signaling in *Megaselia* (*Figure 4—figure supplement 4*), it appears to function independently of the *Mab-doc/doc2*-dependent feedback loop (*Figure 4—figure supplement 5*).

Lastly, we examined the regulation of *Mab-egr* expression. In *Drosophila*, *egr* expression begins at the syncytial blastoderm stage under the control of both BMP signaling and *zen*, whereas in *Megaselia*, *Mab-egr* expression begins at the onset of gastrulation. In *Mab-dpp* knockdown embryos, *Mab-egr* expression was completely absent (*Figure 4E–F*). In *Mab-doc/doc2* knockdown embryos, *Mab-egr* expression was greatly reduced (*Figure 4G*). *Mab-doc/doc2/hnt* triple knockdown did not further reduce *Mab-egr* expression during gastrulation (not shown). *Mab-zen* knockdown embryos displayed only a slight reduction in *Mab-egr* expression during gastrulation; however, at germ band extension, *Mab-zen* knockdown embryos displayed an increase in the number of *Mab-egr* expressing cells due to the transformation of serosa into amnion (*Figure 4H–I*). Thus, in *Megaselia*, *Mab-egr* is primarily under the control of *Mab-doc*, not *Mab-zen*.

## Conclusions

While previous work has demonstrated that BMP gradients can form and be stabilized through molecular feedback circuits (*Bier and De Robertis, 2015*), we have shown here that positive feedback circuits can also be an important target in the evolution of body plans. Specifically, the distinct BMP gradients of *Megaselia* and *Drosophila*, which result in the specification of distinct extraembryonic tissue complements, are the result of spatiotemporal changes in an *egr*-dependent positive feedback circuit (*Figure 4J*).

Given that *Megaselia* has retained distinct serosa and amnion tissues, the function of the BMP gradient in this species might be similar to the ancestral condition in higher flies. What changes to the underlying genetic network during evolution would have been necessary to transform the shape of the *Megaselia* BMP gradient into that seen in *Drosophila*? In blastoderm embryos of *Megaselia*, the BMP gradient establishes the expression of *Mab-zen* in prospective serosa tissue, and *Mab-doc/doc2* and *Mab-hnt* in prospective serosa and amnion tissues. While this patterning phase is not sufficient to differentiate between serosa and amnion tissue, it sets the stage for *Mab-doc/doc2*-dependent *Mab-egr* expression during gastrulation. The *Drosophila* gene network that regulates *egr* expression differs at least in two ways. First, *doc* (along with *hnt*) is expressed downstream of *zen*. Conceptually, this regulatory difference is sufficient to explain the difference of *egr* expression between the two species during gastrulation, and hence also the difference in BMP signaling and tissue specification. We therefore propose that this change led the evolutionary transition. Once *doc*

was downstream of *zen*, the latter might have gradually gained control over *egr*. This scenario is consistent with our observation that even in *Megaselia*, *Mab-zen* slightly promotes *Mab-egr* expression. Second, *Drosophila* acquired a BMP-independent *zen* expression domain in the syncytial blastoderm, which is not observed in other dipterans (*Goltsev et al., 2007*; *Rafiqi et al., 2008*). The acquisition of this early *zen* domain could have promoted *egr* expression in the blastoderm of *Drosophila*, where *egr* is part of a *zen*-dependent network that confers robustness to the BMP gradient (*Gavin-Smyth et al., 2013*).

Our data suggest that the ancestral function of the positive feedback circuit was to promote amnion specification through BMP signaling. While the identity of regulatory factors of the positive feedback circuit may be evolutionarily labile (in Tribolium *doc* and *hnt* appear to be dispensable for amnion specification [*Horn and Panfilio, 2016*]), we propose that the mechanism of amnion specification through feedback-driven BMP signaling dynamics applies to a wide range of insects, because in Tribolium the pMad domain also gradually shifts to become elevated in the presumptive amnion during early gastrula stages (*van der Zee et al., 2006*; *Nunes da Fonseca et al., 2008*). The principle of evolving tissue complexity through changes in positive feedback circuits of morphogen gradients has not yet been documented in other developmental contexts, but it may also apply to unrelated traits, such as eyespots on butterfly wings (*Monteiro, 2015*).

## Materials and methods

### dsRNA and mRNA synthesis, injections and fixation

RNAi was performed as described (*Rafiqi et al., 2008*, *2010*). In negative controls, embryos were injected with dsRNA against the *huckebein* homolog from another fly species, *Eba-hkb* (*Lemke et al., 2010*). dsRNAs for *Eba-hkb*, *Mab-hnt*, *Mab-zen*, and *Mab-dpp* were prepared as described (*Rafiqi et al., 2008*; *Lemke et al., 2010*; *Rafiqi et al., 2010*). The following primers were used to synthesize dsRNA against *Mab-doc*: (5'-CCAAGCCTTC<u>TAATACGACTCACTATAGGGAGA</u>GACGAGGATGGCGAGTACTG-3' and 5'-CAGAGATGCA<u>TAATACGACTCACTATAGGGAGA</u>GTTCCCACCAATGGTTGTGC-3'), *Mab-doc2*: (5'-CCAAGCCTTC<u>TAATACGACTCACTATAGGGAGA</u>TGAGTGGTGTGGATATCGCG-3' and 5'-CAGAGATGCA<u>TAATACGACTCACTATAGGGAGA</u>AGCAAGGACAGTGTGACCAT-3'), *Mab-cv-2*: (5'-CAGAGATGCA<u>TAATACGACTCACTATAGGGAGA</u>ACGGCGCAAATCCGACTGTTGT-3' and 5'-CCAAGCCTTC<u>TAATACGACTCACTATAGGGAGA</u>AACGCAGAGTGGAGCCGCTT-3'), and *Mab-egr*: (5'-CGCCGCGGTCTACATCACTGTT-3' and 5'-CGCCGCGGTCTACATCACTGTT-3'). T7 promoters are underlined. To create the template for capped *Mab-doc, Mab-hnt, Mab-dpp,* and *Mab-eve* mRNAs, complete ORFs were PCR-amplified from embryonic cDNA using primers with attB recombination sites attached at the 5' ends. The following primers were used: *Mab-hnt* (5'-GGGG<u>ACAAGTTTGTACAAAAAAGCAGGCT</u>ACCATGCTTCATGCAACCAACC-3' and 5'-GGGG<u>ACCACTTTGTACAAGAAAGCTGGGT</u>CTACTTCTCAACACCCAAGAACTTG-3'), *Mab-doc* (5'-GGGG<u>ACAAGTTTGTACAAAAAAGCAGGCT</u>AAAATGATTACCATGAATGAATTAGTG-3' and 5'-GGGG<u>ACCACTTTGTACAAGAAAGCTGGGT</u>CTAACATTGCGCAACACCCAAAA-3'), *Mab-dpp* (5'-GGGG<u>ACAAGTTTGTACAAAAAAGCAGGCT</u>AAAATGCGCGCATGGCTT-3' and 5'-GGGG<u>ACCACTTTGTACAAGAAAGCTGGGT</u>TCATCGACATCCACATCCAAC-3'), and *Mab-eve* (5'-GGGG<u>ACAAGTTTGTACAAAAAAGCAGGCT</u>AAAATGCAAGGATACAGAAACTACA-3' and 5'-GGGG<u>ACCACTTTGTACAAGAAAGCTGGGT</u>TTAGGCCTCACTCTCTGTCTT-3'). The attB recombination sites are underlined. The ORFs were cloned into a destination vector which was modified from the pSP35T (*Amaya et al., 1991*) using Gateway Cloning (Life Technology). Capped mRNAs were prepared using SP6 polymerase with the mMessage mMachine Kit (Ambion). Sequence information of *Mab-doc2* (KY302676), *Mab-egr* (KY302677), and *Mab-cv-2* (KY302678) is provided in Genbank. For microinjection, embryos were collected and aligned on a glass slide along a 0.2 mm glass capillary, briefly desiccated, and covered with halocarbon oil (Sigma H773) at room temperature as described (*Rafiqi et al., 2011*). Stages of the embryos were defined according to (*Wotton et al., 2014*). Embryos were injected before the syncytial blastoderm stage (~1:30–2:30 hr at 18°C after egg deposition) unless otherwise specified. Injected embryos were then heat fixed and manually devitellinized as described (*Rafiqi et al., 2011*) before in situ hybridization and immunostaining.

## RNA in situ hybridization, immunohistochemistry and image analysis

RNA probes were labeled with digoxigenin (*Mab-egr*, *Mab-dpp*), fluorescein (*Mab-doc*, *Mab-doc2*, *Mab-zen* and *Mab-hnt*) and biotin (*Mab-eve*) as described (*Tautz and Pfeifle, 1989*; *Kosman, 2004*). Probe templates for *Mab-eve*, *Mab-zen*, and *Mab-hnt* were synthesized as described (*Bullock et al., 2004*; *Rafiqi et al., 2008*, *2010*). Other probes were synthesized from PCR templates obtained from an embryonic cDNA library using the following primers: *Mab-egr*5' (5'-CCAAGCCTTC<u>TAATACGAC TCACTATAGGG</u>GAGATGAGCTGCTGCCAGAGCGTT-3' and 5'-CAGAGATGCA<u>ATTAACCCTCAC TAAAGGG</u>AGATGTGCATTTTGTGATTATTGAAAGT-3'), *Mab-egr3'* (5'-CCAAGCCTTC<u>TAATACGAC TCACTATAGGG</u>GAGAAACTATGAGACAAATACTTAACGGA-3' and 5'-CAGAGATGCA<u>ATTAACCC TCACTAAAGGG</u>AGATCGAGCGATTGACGTCTCAGT-3'), *Mab-doc* (5'-CCAAGCCTTC<u>TAATACGAC TCACTATAGGG</u>GAGAGACGAGGATGGCGAGTACTG-3' and 5'-CAGAGATGCA<u>ATTAACCCTCAC TAAAGGG</u>AGAGTTCCCACCAATGGTTGTGC-3'), *Mab-doc2* (5'-CCAAGCCTTC<u>ATTAACCCTCAC TAAAGGG</u>AGATGAGTGGTGTGGATATCGCG-3' and 5'-CCAAGCCTTC<u>TAATACGACTCACTA TAGGG</u>AGATGAGTGGTGTGGATATCGCG-3'), *Mab-cv-2* (5'-CAGAGATGCA<u>ATTAACCCTCAC TAAAGGG</u>AGAACGGCGCAAATCCGACTGTTGT-3' and 5'-CCAAGCCTTC<u>TAATACGACTCACTA TAGGG</u>AGAAACGCAGAGTGGAGCCGCTT-3') and the synthesis of other probes were previously described (*Rafiqi et al., 2008*, *2010*). T3 and T7 promoters are underlined in the forward and reverse primers, respectively. The following procedures for RNA in situ hybridization and immunostaining were done as described (*Rafiqi et al., 2012*). For RNA in situ hybridization, *Megaselia* embryos were heat fixed, while for immunostaining, they were fixed by formaldehyde except for quantification (see below). pMad was detected with a rabbit monoclonal antibody against Smad3 phosphorylated on Serine 423 and Serine 425 (Epitomics, Cat# 1880–1) at 1:250 dilution. For two-color fluorescent in situ hybridization, confocal scans were done with a Zeiss LSM510 laser-scanning microscope. All subsequent image quantification and analysis of confocal micrographs were done in ImageJ (*Schneider et al., 2012*). To quantify pMad staining intensity, embryos were stained with pMad as described (*Rafiqi et al., 2012*) after a modified fixation protocol. To preserve better morphology for quantification, heat fix was used instead of formaldehyde. The embryos were treated with a boiling solution of 0.7% NaCl and 0.05% Triton X-100 followed by a heptane and methanol devitellinization step. Postfixation was done with 5% formaldehyde in a 3:1 mixture of phosphate buffered saline (PBT; 137 mM NaCl, 2.7 mM KCl, 10 mM $Na_2HPO_4$, 2 mM $KH_2PO_4$, 0.1% Triton X-100 pH 7.4) and methanol. This was followed by a second heptane and methanol devitellinization step. Embryos at early gastrulation were staged after the initiation of cephalic furrow and before the dorsal-most point of the proctodeum reached 20% of total egg length. The quantification of pMad staining in injected *Megaselia* embryos followed the *Drosophila* protocol (*Gavin-Smyth et al., 2013*). To compare whether there was a significant reduction of pMad intensity in *Mab-egr* RNAi compared to wild type, a Wilcoxon rank sum test was performed in R [R Core Team (2012). R: A language and environment for statistical computing. R Foundation for Statistical Computing, Vienna, Austria. http://www.R-project.org].

## Gene trees

Amino acid sequences of *Dorsocross* homologs in different species with the following reference numbers were retrieved (*Aedes aegypti*; *Aae-docA* XP_001648597.1 and *Aae-docB* XP_001663692.1), (*Anopheles gambiae*; *Aga-docA* XP_315924.3 and *Aga-docB* EAA11871.5), (*Drosophila melanogaster*; *Doc1* AAF50328.2, *Doc2* AAF50329.1 and *Doc3* AAF50331.1), (*Drosophila pseudoobscura*; *Dps-Doc1* EAL31211.1, *Dps-Doc2* EAL31212.2, and *Dps-Doc3* EAL31210.1), (*Drosophila grimshawi*; *Dgr-Doc1* EDV96918.1, *Dgr-Doc2* EDV96917.1 and *Dgr-Doc3* EDV96915.1). Full-length protein alignments were created using the MUSCLE program with default parameters (*Edgar, 2004*). The best amino acid substitution model was estimated using AIC in ProtTest 3 (*Darriba et al., 2011*) and the LG model was chosen. Maximum likelihood trees were calculated using PhyML 3 (*Criscuolo, 2011*). Bootstrap values were based on 1000 replicas.

## Acknowledgements

We thank Jeff Klomp (University of Chicago) and Steffen Lemke (University of Heidelberg) for technical advice and discussions, and Sally Horne-Badovinac (University of Chicago) and Robert Ho

(University of Chicago) for comments on the manuscript. This work was supported by the National Science Foundation Grant IOS-1121211 to US-O and an award of the University of Chicago Hinds Fund to CWK.

## Additional information

### Funding

| Funder | Grant reference number | Author |
|--------|------------------------|--------|
| National Science Foundation | IOS-1121211 | Urs Schmidt-Ott |
| University of Chicago | Hinds Fund graduate student award | Chun Wai Kwan |

The funders had no role in study design, data collection and interpretation, or the decision to submit the work for publication.

### Author contributions

CWK, Conception and design, Acquisition of data, Analysis and interpretation of data, Drafting or revising the article; JG-S, ELF, Analysis and interpretation of data, Drafting or revising the article; US-O, Conception and design, Analysis and interpretation of data, Drafting or revising the article

### Author ORCIDs

Urs Schmidt-Ott, http://orcid.org/0000-0002-1351-9472

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
