## [Decision Letter]

Thank you for submitting your article "Functional evolution of a morphogenetic gradient" for consideration by *eLife*. Your article has been reviewed by two peer reviewers, and the evaluation has been overseen by Marianne Bronner as the Senior Editor and Reviewing Editor. The reviewers have opted to remain anonymous.

The reviewers have discussed the reviews with one another and the Reviewing Editor has drafted this decision to help you prepare a revised submission.

Summary:

Kwan et al. investigate BMP patterning in embryos of the fly *Megaselia* and compare this with known mechanisms acting in *Drosophila*. Data support the view that evolutionary changes are responsible for differences in tissue specification found within these two fly species. In *Megaselia*, serosa and amnion form as two distinct tissues, whereas the aminoserosa forms as one merged tissue in *Drosophila*. The data, composed mostly of RNAi, ectopic expression, and colocalization of BMP ligands and target genes, makes a compelling case for differences in feedback control of BMP signaling between these two species as causative for these tissue differences. Specifically, the data suggest that the gene networks acting in these two fly species have changed with regard to the regulation of the gene *eiger (egr*). In *Drosophila*, prior published studies have demonstrated *egr* is a positive regulator of BMP signaling, acting downstream of *decapentaplegic (dpp*) and *zerknüllt (zen*) genes. Here, the authors show that in *Megaselia, egr* also acts downstream of *dpp* but, in contrast to the case in *Drosophila*, that in *Megaselia egr* is not regulated by *zen* (or rather, it has a minor role). Instead, in *Megaselia, egr* is regulated by the *dorsocross* genes (*doc/doc2*). This change in regulation of *egr* is proposed as the mechanism by which amnion regulation (through *doc* genes and *hnt*) was separated from serosa regulation (mediated by *zen*).

The results are interesting, novel and well controlled and the manuscript is well written. Some additional experiments are required as outlined below.

Essential revisions:

1) Why does *Mab-egr* RNAi reduce pMad levels uniformly but not the width of the pMad stripe; whereas, *Mab-doc* genes RNAi reduces pMad levels and the width of the stripe?

Essentially, pMad within the amnion domain of *Mab-egr* RNAi embryos appears "normal" (stripe is just as wide) whereas in the *Mab-doc* mutants the pMad stripe is thinner. Could this relate to a role of *Mab-doc* genes in regulating a *Mab-cv-2*? Or other target genes besides *egr*?

2) Is there *cv-2* in *Megaselia* and does it also act together with *Mab-egr* to support positive feedback? The authors may not want to address *cv-2* experimentally, but at least a sentence or two should be added to the text to put the current study in context of what's known in *Drosophila*.

3) How certain are the authors that *Mab-zen* RNAi is working? (e.g. Figure 3—figure supplement 1, panels A-C). Is *zen* expression lost at st 5 and st 6? Can *Mab-zen* RNAi lead to efficient decrease in levels earlier to support results of Figure 2 and Figure 3, for example?

4) On the flip side, how certain is it that *Mab-zen* levels are not affected upon *Mab-dpp* RNAi? Levels in Figure 2 do appear reduced relative to Figure 2. Can *zen* levels be quantified? And possibly compared to *zen* levels upon *Mab-zen* RNAi.

---

## [Author Response]

*Essential revisions:*

*1) Why does Mab-egr RNAi reduce pMad levels uniformly but not the width of the pMad stripe; whereas, Mab-doc genes RNAi reduces pMad levels and the width of the stripe?*

*Essentially, pMad within the amnion domain of Mab-egr RNAi embryos appears "normal" (stripe is just as wide) whereas in the Mab-doc mutants the pMad stripe is thinner. Could this relate to a role of Mab-doc genes in regulating a Mab-cv-2? Or other target genes besides egr?*

*2) Is there cv-2 in Megaselia and does it also act together with Mab-egr to support positive feedback? The authors may not want to address cv-2 experimentally, but at least a sentence or two should be added to the text to put the current study in context of what's known in Drosophila.*

We agree that BMP signaling and *Mab-doc/doc2* activity together could regulate more than one gene with a role in shaping the BMP gradient and thank the reviewers for pointing this out. We address this issue in a new paragraph (–subsection “*Mab-doc*-dependent control of *Mab-egr* expression contributes to a positive feedback circuit that promotes BMP signaling during gastrulation”, fourth paragraph) and added a brief description of *cv-2* experiments in *Megaselia* to the revised manuscript (Figure 4—figure supplement 3, Figure 4—figure supplement 4 and Figure 4—figure supplement 5).

For *Drosophila*, Gavin-Smyth et al. (2013) found that the reduction of BMP signaling was less severe in *egr* deficient embryos than in *Medea* deficient embryos that completely lack BMP-dependent positive feedback, suggesting that *egr* is not the only target of BMP signaling involved in the feedback process. Gavin-Smyth et al. (2013) also presented an analysis of the *cv-2* phenotype. In *Drosophila*, the initial expression of *cv-2* is not under BMP control but under the control of *zen*, which is expressed independently of BMP signaling in stage 5 embryos; therefore Gavin-Smyth et al. did not consider it part of the feedback circuitry in *Drosophila*. They also reported that loss of *cv-2* causes an elevation of BMP signaling, indicating it does not simply act as a component in a positive feedback circuit. In the new paragraph of our manuscript, we explicitly acknowledge the possibility of additional BMP/doc target genes acting as feedback components in *Megaselia* and quote the Gavin–Smyth paper for a potential parallel with *Drosophila* where BMP signaling seems to control more than one feedback component.

In *Megaselia, Mab-cv-2* expression is widely expressed from stage 6 onwards, with slightly higher expression dorsally, and was not grossly perturbed by *Mab-doc/doc2* knockdown, suggesting that, like in *Drosophila*, it is not a component of the feedback process. Interestingly, we find that knockdown of *Mab-cv-2* can perturb amnion specification at the beginning of gastrulation, suggesting that *Mab-cv-2* might promote BMP signaling at the beginning of gastrulation, unlike in *Drosophila* where it attenuates BMP signaling. This difference between *Megaselia* and *Drosophila* is not completely surprising because Cv-2 has been shown to act in a context and dose dependent manner to either promote or inhibit BMP signaling. In conclusion, while the only confirmed *Mab-doc/doc2* target remains *Mab-*egr, we have modified the manuscript to take into account the likelihood that there are other target genes.

*3) How certain are the authors that Mab-zen RNAi is working? (e.g. Figure 3—figure supplement 1, panels A-C). Is zen expression lost at st 5 and st 6? Can Mab-zen RNAi lead to efficient decrease in levels earlier to support results of Figure 2 and Figure 3, for example?*

As we have shown previously, *Mab-zen* RNAi is effective within less than 10 minutes after injection of dsRNA (supplemental Figure 1A in Rafiqi et al. 2010). To clarify this issue, we replaced panels of the original Figure 3—figure supplement 1 (showing embryos at stage 8 stained for *Mab-zen, Mab-doc* and *Mab-hnt* following *Mab-zen* RNAi) with four new panels showing representative *Mab-zen* expression patterns at stage 6 in a control-injected embryo (A) and in a *Mab-zen* RNAi embryo (B), in addition to stage-matched *Mab-zen* RNAi embryos stained with *Mab-doc* and *Mab-hnt* probes (C, D). In *Mab-zen* RNAi embryos, only residual punctate *Mab-zen* expression was observed, consistent with high transcriptional activity in the nuclei during gastrulation. In the cytoplasm, *Mab-zen* transcript did not accumulate above a background level, indicating that *Mab-zen* expression is post-transcriptionally knocked down in *Mab-zen* RNAi embryos.

*4) On the flip side, how certain is it that Mab-zen levels are not affected upon Mab-dpp RNAi? Levels in Figure 2 do appear reduced relative to Figure 2. Can zen levels be quantified? And possibly compared to zen levels upon Mab-zen RNAi.*

In the original figure, contrast enhancement differed between the images shown in panels 2H and I. When original pictures of late *Mab-dpp* RNAi embryos were processed in exactly the same way as the controls, *Mab-zen* expression levels were similar between controls and late *Mab-dpp* RNAi embryos (n=6). However, we noticed that the green background caused by the *Mab-eve* probe varied slightly between embryos. In the revised figure, the image shown in 2I was replaced with an image that was subjected to same contrast enhancement that the image shown in 2I.